# Association of Antihypertensive Effects of Esaxerenone with the Internal Sodium Balance in Dahl Salt-Sensitive Hypertensive Rats

**DOI:** 10.3390/ijms23168915

**Published:** 2022-08-10

**Authors:** Mai Hattori, Asadur Rahman, Satoshi Kidoguchi, Nourin Jahan, Yoshihide Fujisawa, Norihiko Morisawa, Hiroyuki Ohsaki, Hideki Kobara, Tsutomu Masaki, Akram Hossain, Akumwami Steeve, Akira Nishiyama

**Affiliations:** 1Department of Pharmacology, Faculty of Medicine, Kagawa University, 1750-1 Ikenobe, Miki-cho, Kita-gun, Kagawa 761-0793, Japan; 2Life Science Research Center, Faculty of Medicine, Kagawa University, 1750-1 Ikenobe, Miki-cho, Kita-gun, Kagawa 761-0793, Japan; 3Department of Medical Biophysics, Kobe University Graduate School of Health Sciences, 7-10-2, Tomogaoka, Suma-ku, Kobe 654-0142, Hyogo, Japan; 4Department of Gastroenterology and Neurology, Faculty of Medicine, Kagawa University, 1750-1 Ikenobe, Miki-cho, Kita-gun, Kagawa 761-0793, Japan; 5Department of Anesthesiology, Faculty of Medicine, Kagawa University, 1750-1 Ikenobe, Miki-cho, Kita-gun, Kagawa 761-0793, Japan

**Keywords:** esaxerenone, nonsteroidal MRB, salt-sensitive hypertension, body sodium

## Abstract

Background: The nonsteroidal mineralocorticoid receptor blocker esaxerenone is effective in reducing blood pressure (BP). Objective: In this study, we investigated esaxerenone-driven sodium homeostasis and its association with changes in BP in Dahl salt-sensitive (DSS) hypertensive rats. Methods: In the different experimental setups, we evaluated BP by a radiotelemetry system, and sodium homeostasis was determined by an approach of sodium intake (food intake) and excretion (urinary excretion) in DSS rats with a low-salt diet (0.3% NaCl), high-salt diet (HSD, 8% NaCl), HSD plus 0.001% esaxerenone (*w*/*w*), and HSD plus 0.05% furosemide. Results: HSD-fed DSS rats showed a dramatic increase in BP with a non-dipper pattern, while esaxerenone treatment, but not furosemide, significantly reduced BP with a dipper pattern. The cumulative sodium excretion in the active period was significantly elevated in esaxerenone- and furosemide-treated rats compared with their HSD-fed counterparts. Sodium content in the skin, skinned carcass, and total body tended to be lower in esaxerenone-treated rats than in their HSD-fed counterparts, while these values were unchanged in furosemide-treated rats. Consistently, sodium balance tended to be reduced in esaxerenone-treated rats during the active period. Histological evaluation showed that esaxerenone, but not furosemide, treatment attenuated glomerulosclerosis, tubulointerstitial fibrosis, and urinary protein excretion induced by high salt loading. Conclusions: Collectively, these findings suggest that an esaxerenone treatment-induced reduction in BP and renoprotection are associated with body sodium homeostasis in salt-loaded DSS rats.

## 1. Introduction

Hypertension is considered a leading cause of cardiovascular disease and is associated with mortality worldwide [1]. Decades of research have shown a causal relationship between hypertension and dietary salt intake. Even though the concept of salt-sensitive hypertension (SSH) has emerged from variable blood pressure (BP) responses to dietary salt among individuals [2], its precise underlying pathophysiological mechanism and treatment regimens remain elusive.

The kidney plays a crucial role in the regulation of systemic sodium and water homeostasis, and thus maintains systemic BP [3]. Excessive dietary intake of sodium promotes the kidney to excrete sodium into the urine to maintain body sodium balance by pressure natriuresis [4]. In cases of impaired renal sodium excretion, dietary salt intake leads to a good proportion of sodium accumulation in the total extracellular space, and thus increases extracellular volume. In the SSH animal model, not only salt-induced volume expansion, but also volume-independent salt accumulation, contribute to the development of hypertension [5]. Therefore, in addition to the maintenance of sodium and water homeostasis by the kidney, the distribution of sodium in different tissues may also be associated with the pathophysiology of hypertension.

In healthy subjects, BP exhibits a diurnal variation with a physiological dipping (>10%) during the nighttime (inactive period, dipper pattern) compared with the daytime (active period). However, in subjects with SSH, BP fails to drop, and this is referred to as the non-dipper pattern [6]. Notably, the non-dipper pattern of BP is considered as an important risk factor for cardiovascular disease [7]. The circadian rhythm of urinary sodium excretion is strongly associated with the diurnal variation in BP, while the night/day ratio of sodium excretion is lower in dippers and higher in non-dippers [8]. Therefore, the daytime urinary excretion of sodium is a determinant of nocturnal BP and dipping [9]. Consequently, the non-dipper pattern of BP could be associated with altered sodium homeostasis in the body.

Aggravated mineralocorticoid receptor (MR) signaling is considered to be the major mechanism of end-organ damage, even with low or normal serum aldosterone concentrations, especially in the context of SSH [10]. Shibata et al. [11] showed that high-salt loading induces renal Rac1 upregulation, which subsequently potentiates MR activation, leading to an elevation in BP, despite low serum aldosterone concentrations. Therefore, MR antagonism is considered to be an effective therapeutic approach in SSH. Indeed, treatment with steroidal MR antagonists has effectively reduced BP in SSH subjects with low circulating aldosterone concentrations [12,13,14]. Because non-specific side effects driven by steroidal MR antagonists remain an issue in the clinical setting [15], newly developed nonsteroidal MR blockers have emerged [16]. Among these, esaxerenone (CS-3150, Daiichi Sankyo) has been developed as a third-generation non-steroidal MR antagonist with greater affinity and selectivity over the steroidal MR antagonists [17,18]. Importantly, esaxerenone is effective in reducing BP in patients with hypertension [19,20]. In the context of SSH, our research team and others have shown that esaxerenone treatment significantly reduces BP and cardiorenal injury in high salt (HS)-loaded Dahl salt-sensitive hypertensive (DSS) rats [21,22,23]. Furthermore, 2-week administration of esaxerenone (3 mg/kg/day) significantly reduced BP in deoxycorticosterone acetate/salt-loaded rats compared with their control counterparts, but not in those with eplerenone and spironolactone loading (30 mg/kg/day) [24].

Surprisingly, information is lacking regarding esaxerenone-induced modulation of body sodium homeostasis and its association with the diurnal variation in BP, especially in subjects with SSH. Therefore, we aimed to investigate the effects of esaxerenone and a conventional diuretic, furosemide, on body sodium homeostasis and renal histological changes, and their association with changes in BP in the active and inactive periods in HSD-fed DSS rats.

## 2. Results

### 2.1. Esaxerenone Treatment Reduces BP in HSD-Fed DSS Rats

HSD-fed DSS rats had significantly higher hourly measurements (most time points) and averaged 24 h mean systolic arterial pressure ((SAP) 215 ± 4 mmHg), mean diastolic arterial pressure ((DAP) 150 ± 2 mmHg), and mean arterial pressure ((MAP) 181 ± 5 mmHg) than LSD-fed DSS rats with feeding for 4 weeks (Figure 1A,B and Appendix A). However, concomitant treatment with esaxerenone significantly suppressed the elevation in hourly measurements at certain time points and averaged 24 h SAP (17 ± 7 mmHg), DAP (122 ± 5 mmHg), and MAP (149 ± 6 mmHg). The supplementation of an HSD with furosemide tended to reduce SAP (204 ± 8 mmHg), DAP (140 ± 7 mmHg), and MAP (170 ± 7 mmHg), but these changes were not significant.

At 10 weeks of age, HSD-fed DSS rats showed significantly higher SAP and MAP in the active (dark) and inactive (light) periods than those fed an LSD (Figure 1C,D). Moreover, mean 12 h SAP and MAP were not significantly different between the active and inactive periods, which suggested that HSD-fed DSS rats had a non-dipper pattern of BP. Treatment with esaxerenone in HSD-fed DSS rats significantly reduced BP in the active and inactive periods, and the reduction in BP in the inactive period was greater than that in the active period. These findings suggested a trend of changing from the non-dipper to the dipper pattern of BP. Furosemide treatment in HSD-fed DSS rats also tended to reduce BP in the active and inactive periods, but there was no obvious difference in BP between the active and inactive periods. Twenty-four-hour heart rate was not altered by any of the interventions (Appendix A).

### 2.2. Esaxerenone Treatment Decreases Urinary Sodium Excretion

The level of urinary excretion of sodium during the first 12 h of the active period was significantly higher in HSD-fed DSS rats than in LSD-fed DSS rats. Concomitant treatment with esaxerenone or furosemide in HSD-fed DSS rats also resulted in significantly higher urinary sodium excretion than that in LSD-fed DSS rats in the active period (Appendix A). Notably, urinary potassium excretion was relatively higher in HSD-fed plus furosemide-treated rats than in HSD-fed plus esaxerenone-treated rats in the active period (Appendix A). Interestingly, the cumulative urinary excretion of sodium in the active period, but not in the inactive period, was significantly higher in HSD-fed plus esaxerenone-treated rats and HSD-fed plus furosemide-treated rats than in HSD-fed DSS rats (Figure 2A–C). In contrast, cumulative urinary excretion of potassium tended to be higher in HSD-fed plus furosemide-treated rats, but it was similar in the HSD-fed and HSD-fed plus esaxerenone-treated rats (Figure 2D–F).

### 2.3. Esaxerenone Treatment Tends to Reduce Skin and Carcass Sodium Concentrations

To investigate whether esaxerenone affects internal solute distribution, water content (amount of water/dry tissue weight), and sodium and potassium concentrations (amount of sodium or potassium/dry tissue weight) were measured in the skin, skinned carcass, and whole carcass. Although there were no significant differences in the wet or dry weight of the skin, skinned carcass, or whole carcass, these variables tended to be higher in HSD-fed plus esaxerenone-treated DSS rats than in the other groups. However, the water content was similar in all tissues among the treatment groups (Appendix A–I). As expected, HSD-fed DSS rats showed significantly higher sodium concentrations in the skin, skinned carcass, and whole carcass than those in LSD-fed rats (Figure 3A,D,G). HSD-fed plus furosemide-treated rats showed significantly higher sodium concentrations in skin compared to the HSD-fed rats, while HSD-fed plus esaxerenone-treated rats showed a reduced trend of sodium concentrations (significantly higher than LSD but not with HSD) in the skin, skinned carcass, and whole carcass. Importantly, level of sodium in the skinned and whole carcass was significantly higher in furosemide-treated rats compared with the esaxerenone-treated rats. Potassium concentrations in the skin and skinned carcass were significantly altered in HSD-fed rats compared to the LSD-fed rats. However, intervention with esaxerenone and furosemide did not affect the potassium level compared to the HSD-fed rats (Figure 3B,E,H). Notably, the sodium/potassium ratio was higher in the skin, but not in the skinned carcass or total body, in HSD-fed rats (Figure 3C,F,I). The addition of furosemide to HSD-fed rats tended to increase the sodium/potassium ratio, while the addition of esaxerenone treatment tended to reduce it. However, at the skinned and whole carcass, the sodium/potassium ratio significantly increased in furosemide-treated rats compared to the esaxerenone-treated rats.

### 2.4. Esaxerenone Treatment Decreases the Sodium Balance in the Active Period

The sodium balance was estimated on the basis of sodium intake (calculated from food intake) and sodium output (through urine), and subsequent normalization with body weight. We found that during the first week of the treatment regimen, sodium balance in the active period tended to be higher in HSD-fed DSS rats than in HSD-fed plus esaxerenone-treated rats and HSD-fed plus furosemide-treated rats. However, in the subsequent weeks, sodium balance was similar in HSD-fed DSS rats and HSD-fed plus furosemide-treated rats, but the sodium balance appeared to show a downward trend in HSD-fed plus esaxerenone-treated rats in the active period (Figure 4A). Because food intake was low during the inactive period, the sodium balance was not obviously different among the groups (Figure 4B). Importantly, the 24 h sodium balance was significantly lower in HSD-fed plus esaxerenone-treated rats at week 4 (Figure 4C).

### 2.5. Esaxerenone Treatment Improves Renal Histology in HSD-Fed DSS Rats

Hematoxylin and eosin staining showed that glomerular and tubular structure was damaged in HSD-fed DSS rats with protein cast formation (Figure 5). However, esaxerenone treatment greatly attenuated the renal histological architecture in HSD-fed DSS rats. LSD-fed DSS rats showed normal glomeruli or slight glomerular injury during the observation period. In contrast, HSD feeding for 4 weeks led to glomerulosclerosis, as evaluated by an increase in the glomerular PAS-positive area. Concomitant treatment with esaxerenone, but not furosemide, caused a significant reduction in the glomerular-PAS positive area. Furthermore, severe tubulointerstitial fibrosis was obvious by Masson trichrome staining in HSD-fed DSS rats compared with age-matched LSD-fed rats. Notably, esaxerenone treatment almost completely attenuated the development of tubulointerstitial fibrosis, but only a tendency for a decrease in the development of tubulointerstitial fibrosis was observed in the furosemide treatment group.

### 2.6. Esaxerenone Treatment Reduces Urinary Protein Excretion

HSD-fed DSS rats showed considerably higher urinary protein excretion in the active period, while a moderate, but still significantly higher protein excretion, was observed in the inactive period than in LSD-fed DSS rats (Figure 6A,B). Esaxerenone treatment significantly reduced the level of urinary protein excretion in the active and inactive periods. However, furosemide reduced the level of urinary protein excretion only in the active period. At 24 h level, both esaxerenone and furosemide intervention significantly reduced the urinary protein excretion compared to the HSD-fed rats (Figure 6C).t 2

## 3. Discussion

An elevation in BP with an altered dipping pattern in response to high dietary salt intake is a characteristic phenomenon of SSH, which enhances cardiorenal morbidity and mortality [6]. We previously showed the antihypertensive effects of a nonsteroidal mineralocorticoid receptor blocker, esaxerenone, in DSS rats [21]. In this study, we examined the effects of esaxerenone and furosemide on the diurnal variation in BP, as well as the associated possible underlying mechanism. We found that treatment with esaxerenone, but not furosemide, not only suppressed the HSD-induced increase in BP, but also changed the diurnal variation in BP from the non-dipper to the dipper pattern. Additionally, esaxerenone treatment caused an increased urinary sodium/potassium ratio with a concomitant decrease in whole-body sodium concentrations, thereby reducing the whole-body sodium balance. Furthermore, a renoprotective effect of esaxerenone was observed in HSD-fed DSS rats, while it was not seen in HSD-fed plus furosemide-treated DSS rats. These findings suggest that an esaxerenone-induced reduction in BP with an improvement in diurnal variation and renoprotection are associated with an improvement in whole-body sodium homeostasis in salt-loaded DSS rats.

Clinical evidence supports a close relationship between SSH and the non-dipper pattern of BP [6]. We previously reported that salt-loaded DSS rats showed a non-dipper pattern of BP [25]. Interestingly, a recent post hoc analysis [26] of the ESAX-HTN study [19] demonstrated the beneficial effects of esaxerenone in older patients with the non-dipper pattern of nocturnal BP. In this study, we found that treatment with esaxerenone attenuated the HSD-induced increase in BP, which was associated with a restored circadian rhythm of BP from a non-dipper to a dipper pattern. Even though a trend of a reduction in BP was observed (not significant) by treatment with furosemide, it did not affect the HSD-induced non-dipper pattern of BP.

Various renal and extra-renal mechanisms are thought to play a role in the non-dipper pattern of BP in SSH. In patients with SSH and chronic kidney disease, altered sodium excretory capability is associated with the diurnal variation in BP [27]. Following HS loading, a change in sodium excretory capacity becomes evident, resulting in escalated nighttime BP, which may contribute to the non-dipper type of BP. In non-dipper patients, diminished natriuresis compensates during the daytime (active period) and augments pressure natriuresis during the nighttime (inactive period) [6]. Accordingly, sodium restriction [28] may stabilize the diurnal variation in BP from the non-dipper to the dipper pattern. In this study, we found that cumulative urinary sodium excretion was significantly increased during the active period in the esaxerenone- and furosemide-treated groups. However, cumulative urinary excretion of potassium tended to be higher in the furosemide-treated group than in the esaxerenone-treated group. Therefore, the urinary sodium to potassium ratio was significantly higher in HSD-fed plus esaxerenone-treated rats. A similar observation was demonstrated in a previous study, which used bilateral adrenalectomized rats and showed an increased sodium to potassium ratio following treatment with esaxerenone [24]. Moreover, finerenone, which is another non-steroidal MRB, dose dependently showed natriuretic potency in healthy volunteers with a fludrocortisone challenge [29]. These collective data suggest that the esaxerenone-induced increase in the urinary sodium to potassium ratio contributes to the improvement in BP, at least in part, in HS-loaded DSS rats. Further studies are required to determine the precise mechanism of the improvement in diurnal variation of BP by esaxerenone.

HS loading in DSS rats causes an increase in total body sodium concentrations accompanied by an increase in bone sodium content and total body water. Importantly, the increase in total body sodium concentrations and water are positively correlated with an increase in MAP [5]. In agreement with these data, we found a significant increase in sodium concentrations in the skin, skinned carcass, and whole body in HSD-fed DSS rats. Interestingly, a decline in sodium concentrations in all these compartments and total body was observed following treatment with esaxerenone in HS-loaded DSS rats. In contrast, treatment with furosemide showed an increase in sodium concentrations in the skin, skinned carcass, and total body. However, potassium concentrations were not different among the groups, and the sodium to potassium ratio was not different in the tissue compartments or total body level among the groups. The sodium balance tended to be reduced, but was not significant, in the active period. However, the 24 h sodium balance was significantly lower in esaxerenone-treated rats. A limitation of this study is that a relatively small number of rats was used for measuring the sodium balance. In this study, we did not observe any change in the water content any of the intervention groups. These data indicate that, at least in part, esaxerenone-induced, but not furosemide-induced, modification of body sodium homeostasis might be associated with the improvement in diurnal variation of BP.

In a previous study, the prevalence of the non-dipper status was inversely correlated with renal function, and an elevated proportion of non-dipper cases was associated with an increasing stage of worsening chronic kidney disease [30]. Moreover, glomerulosclerosis and tubulointerstitial fibrosis are thought to be the most pertinent renal histological parameters associated with nocturnal hypertension [31]. Our previous study also showed that the degree of renal injury by glomerulosclerosis or tubulointerstitial fibrosis was closely associated with the degree of the non-dipping pattern of BP in salt-loaded DSS rats [26]. In agreement with a previous report, the present study showed that treatment with esaxerenone, but not furosemide, preserved renal histological architecture with a concomitant reduction in glomerulosclerosis and tubulointerstitial fibrosis [23]. Furthermore, patients with the non-dipper pattern show a greater level of urinary albumin excretion than those with the dipper pattern [22]. In the present study, we found that 4 weeks of HS loading triggered a sharp increase in proteinuria, while esaxerenone treatment significantly reduced urinary protein excretion in the active and inactive periods in DSS rats. In contrast, furosemide treatment reduced urinary protein excretion in the active but not inactive period compared to the HSD-fed rats. This might be due to the short duration of action that seems a limitation for this study. Taken together, our data suggest that esaxerenone-induced improvement in renal histological changes and reduced proteinuria are associated with a reduction in BP.

In conclusion, this study shows that esaxerenone treatment significantly reduces BP and recovers the diurnal variation in BP from the non-dipper to dipper pattern in HS-loaded DSS rats. Importantly, these beneficial effects are driven by the improvement in whole-body sodium homeostasis and protective effects on the kidneys. These observations may help to expand the understanding of the underlying mechanisms associated with the beneficial effects of esaxerenone, especially in patients with salt-dependent hypertension.

## 4. Materials and Methods

### 4.1. Experimental Animals

The experimental protocols (Protocol No. 18627) and procedures were approved by the Animal Research Committee of Kagawa University (Kagawa, Japan). Five-week-old male DSS rats (Japan SLC, Inc., Shizuoka, Japan) were maintained with standard chow (0.5% NaCl) and an ad libitum water supply. The rats were housed in a specific pathogen-free facility under a controlled temperature (24 °C ± 2 °C) and humidity (55% ± 5%) with a 12 h light–dark cycle for 1 week.

### 4.2. Drugs and Modified Chow

The non-steroidal mineralocorticoid receptor blocker esaxerenone (CS-3150) was provided by Daiichi-Sankyo Co., Ltd. (Tokyo, Japan). Furosemide was purchased from Sigma-Aldrich, Inc. (Saint Louis, MO, USA). Modified chows with a low-salt diet ((LSD) 0.3% NaCl), high-salt diet ((HSD) 8% NaCl), HSD with 0.001% esaxerenone (*w*/*w*), and HSD with 0.05% furosemide (*w*/*w*) were obtained from Oriental Yeast Co., Ltd. (Tokyo, Japan). The concentrations of esaxerenone [22] and furosemide [32] in the HSD were calculated on the basis of previous reports. DSS rats were treated daily with esaxerenone and furosemide at doses of approximately 1 and 40 mg/kg body weight, respectively.

### 4.3. Protocols

At 6 weeks of age, DSS rats were provided an LSD, HSD, HSD with esaxerenone, or HSD with furosemide. The experiments were conducted in three phases with different sets of rats to determine BP by a radiotelemetry system or the tail cuff method, analyze whole-body sodium homeostasis, and obtain tissue and plasma samples for analyzing various parameters, as described below.

#### 4.3.1. Protocol I: BP Measurement with a Radiotelemetry System

To measure BP continuously in conscious rats, a radiotelemetry catheter (Data Science International (DSI), Saint Paul, MN, USA) was inserted into the abdominal aorta via the right femoral artery at 5 weeks of age, as described previously [33]. A radiofrequency transmitter (PA-C40; DSI) and a receiver (RPC-1; DSI) were used to measure BP and heart rate. Data were collected and analyzed using Dataquest ART version 4.3 (DSI) and Power Lab (PowerLab 8/30; AD Instruments), respectively.

At 6 weeks of age, a continuous 24 h baseline BP was measured, and the rats were divided into different intervention groups (*n* = 5) as explained above. After 4 weeks of intervention, at 10 weeks of age, 24 h BP was measured. Moreover, we calculated the 12 h dark or active period (18:00–05:00 h) and 12 h light or inactive period (06:00–17:00 h) BP to evaluate the circadian rhythm.

#### 4.3.2. Protocol II: Analysis of Sodium Homeostasis

A separate set of rats was used to observe the time-dependent changes in urinary sodium excretion and body sodium concentrations. Briefly, at 6 weeks of age, DSS rats were placed in a metabolic cage, and after a 12 h acclimatization period, urine samples were collected every 12 h for the first 2 days that were regarded as the baseline. On the basis of the normalized urinary sodium data, DSS rats were divided into the different intervention groups (*n* = 5–8). Immediately after starting the intervention, the water intake, food intake, and volume of urine samples were measured and collected every 12 h for the first 7 days (week 6) continuously, and then at one time point (12 h dark period and 12 h light period) at every week until week 10. After completing the urine collection, the rats were euthanized to collect the skin and the whole carcass to measure sodium concentrations.

##### Urine Electrolytes, Protein, and Osmolality

Urine sodium and potassium concentrations were measured with an automated analyzer (7020-Automatic Analyzer; Hitachi-High-Technologies Corporation, Tokyo, Japan). The sodium balance was analyzed on the basis of the intake of sodium calculated from the food intake and excretion with the urine during 12 h, and this was normalized by the body weight. Urinary protein and creatinine concentrations were measured using commercial assay kits (Wako Pure Chemical Industries, Ltd., Osaka, Japan). Urine osmolality was measured by vapor pressure osmometry (Wescor).

##### Skin and Carcass Water Content, and Electrolytes

Skin and skinned carcass samples were dried at 90 °C for 72 h in accordance with a previous report [5]. Skin and skinned carcass water content was calculated from the difference between the wet weight and the dry weight. Dried samples were ashed at a maximum of 450 °C for 92 h and then at 600 °C for 42 h. The skin ashes were dissolved in 20 mL of 10% HNO3, while the carcass ashes were dissolved in 50 mL of 5% HNO3. Subsequently, sodium and potassium concentrations of these samples were measured with an atomic absorption spectrometer (AA-7000; Shimadzu, Kyoto, Japan). The sum of skin and skinned carcass data was considered as total body sodium or potassium concentrations.

#### 4.3.3. Protocol III: Renal Histological Examination

At 6 weeks of age, the rats were divided into different intervention groups (*n* = 8) on the basis of the baseline BP (tail cuff method). Body weight and BP were then measured every week. At 10 weeks of age, all rats were euthanized, and kidney samples were collected for histological analysis.

##### Histological Analysis

Renal tissues were dissected and fixed with 10% buffered paraformaldehyde, embedded in paraffin, and sectioned into 3 μm-thick slices. The sections were then stained with hematoxylin and eosin, periodic acid–Schiff (PAS), or Masson trichrome reagent to evaluate global architecture, and glomerular and tubulointerstitial lesions, respectively. The percentage of the PAS-positive area in each experimental group was measured using image analysis software (Image J; NIH, Bethesda, MD, USA). A total of 30−35 glomeruli were examined for each rat, and the average percentage of the affected lesions was calculated for each rat. The extent of the interstitial fibrotic area was quantitatively evaluated using the image analysis software, which determined the area occupied by interstitial tissue positive for Masson trichrome staining. All morphometric measurements were performed in a blinded manner to avoid any bias.

### 4.4. Statistical Analysis

Data are presented as the mean ± SEM. We used a one-way analysis of variance followed by the Newman–Keuls multiple comparison test for all cross-sectional, one-factor data to compare values in the LSD-fed DSS rats with those treated with HSD alone or with concomitant esaxerenone/furosemide treatment. The longitudinal data were analyzed by two-way analysis of variance followed by the Bonferroni post hoc test. A value of *p* < 0.05 was considered statistically significant.

## Figures and Tables

**Figure 1 ijms-23-08915-f001:**
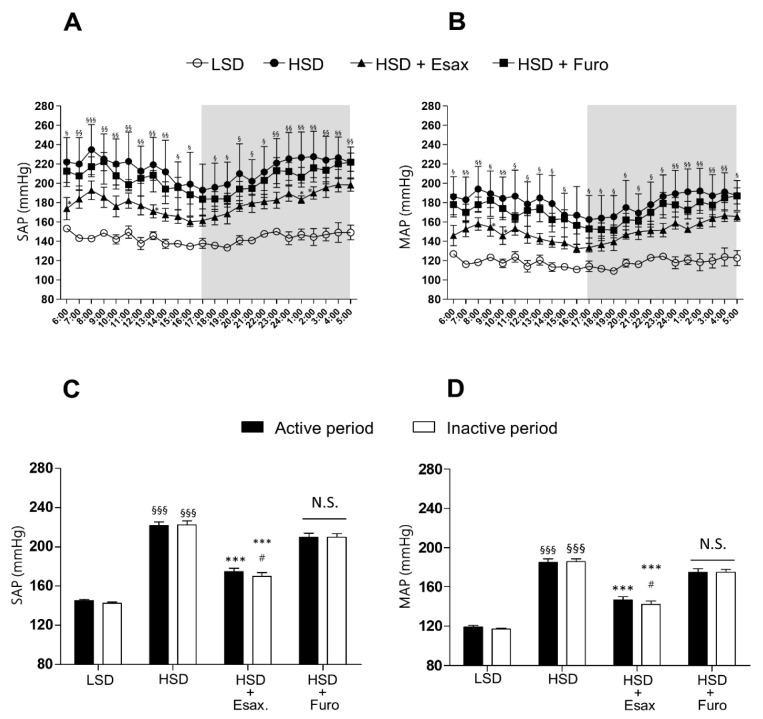
Changes in BP in HS-loaded DSS rats. Twenty-four-hour (**A**) SAP and (**B**) MAP measured by a radiotelemetry system in conscious rats after 4 weeks of treatment with an LSD (*n* = 3), HSD (*n* = 5), HSD plus esaxerenone (*n* = 5), or HSD plus furosemide (*n* = 5). Mean 12 h (**C**) SAP and (**D**) MAP in the active and inactive periods. ^§^
*p* < 0.05, ^§§^
*p* < 0.01, ^§§§^
*p* < 0.001 vs. LSD (in the active and inactive periods); * *p* < 0.05, *** *p* < 0.001 vs. HSD (in the active and inactive periods); ^#^
*p* < 0.05 vs. HSD plus esaxerenone in the active period. N.S., not significant.

**Figure 2 ijms-23-08915-f002:**
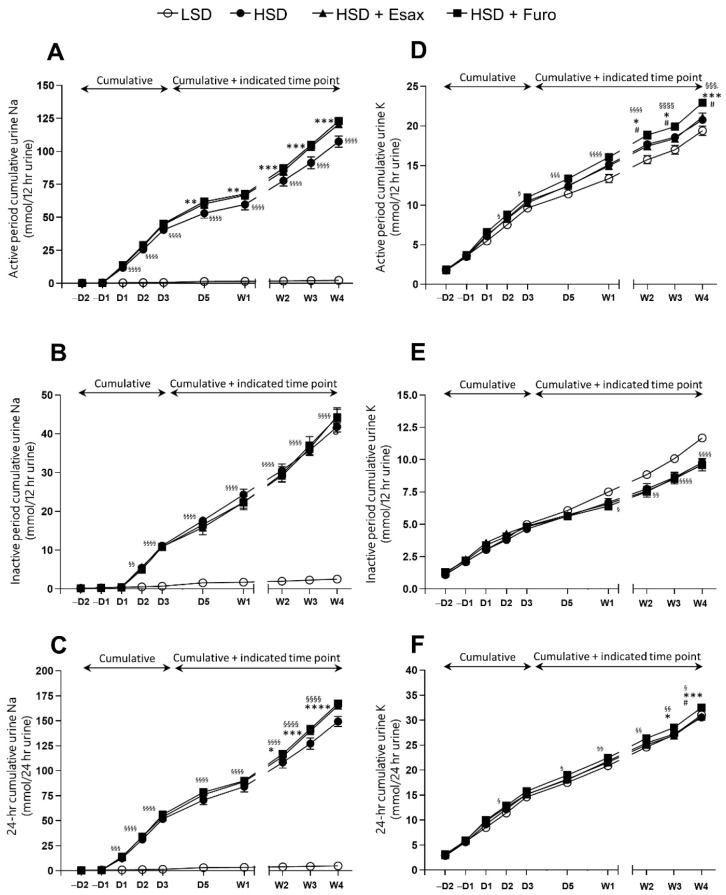
Urinary excretion of Na and K. Cumulative 12 h urinary excretion of Na in the (**A**) active and (**B**) inactive periods and (**C**) at 24 h. Urinary excretion of K in the (**D**) active and (**E**) inactive periods, and (**F**) at 24 h. ^§^
*p* < 0.05, ^§§^
*p* < 0.01, ^§§§^
*p* < 0.001 ^§§§§^
*p* < 0.0001 vs. LSD; * *p* < 0.05, ** *p* < 0.01, *** *p* < 0.001, **** *p* < 0.0001 vs. HSD; ^#^*p* < 0.05 vs. HSD plus esaxerenone.

**Figure 3 ijms-23-08915-f003:**
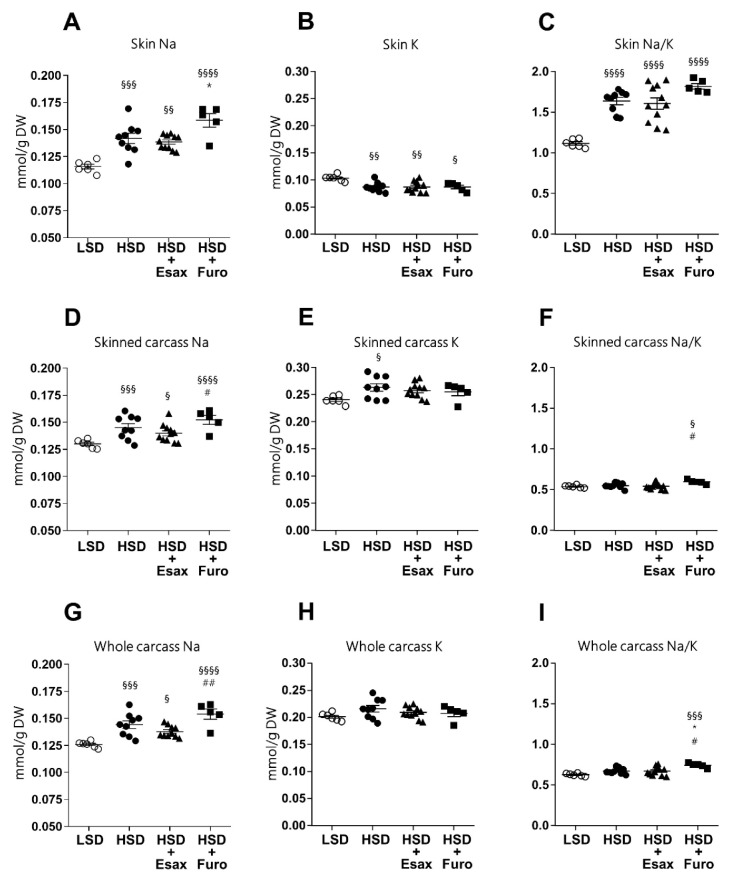
Skin, skinned carcass, and total body Na and K content. Skin (**A**) Na, (**B**) K, and (**C**) the Na/K ratio in the different intervention groups. Skinned carcass (**D**) Na, (**E**) K, and (**F**) the Na/K ratio in the different intervention groups. Total body (**G**) Na, (**H**) K, and (**I**) the Na/K ratio in the different intervention groups. ^§^
*p* < 0.05, ^§§^
*p* < 0.01, ^§§§^
*p* < 0.001, ^§§§§^
*p* < 0.0001 vs. an LSD; * *p* < 0.05, vs. HSD; ^#^
*p* < 0.05, ^##^
*p* < 0.01 vs. HSD plus esaxerenone.

**Figure 4 ijms-23-08915-f004:**
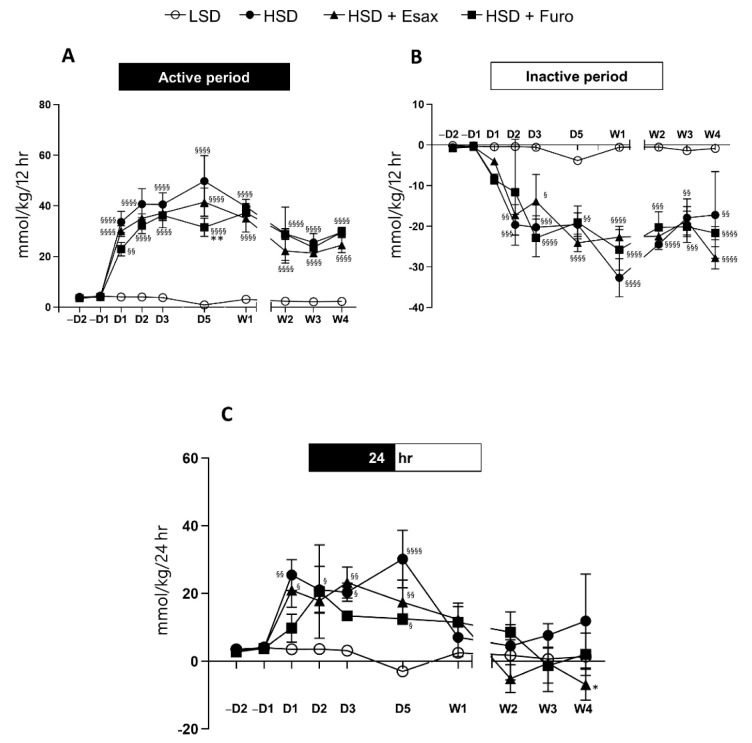
Time-dependent sodium balance. Calculated sodium balance in the (**A**) active and (**B**) inactive periods, and (**C**) at 24 h. ^§^
*p* < 0.05, ^§§^
*p* < 0.01, ^§§§^
*p* < 0.001 ^§§§§^
*p* < 0.0001 vs. LSD; * *p* < 0.05, ** *p* < 0.01 vs. HSD.

**Figure 5 ijms-23-08915-f005:**
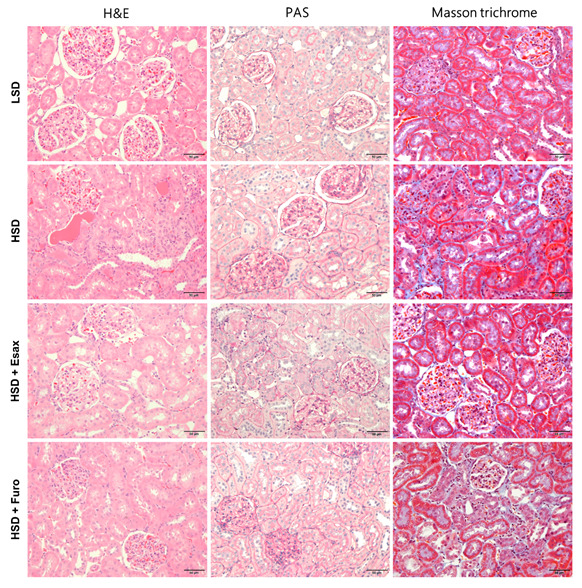
Renal histological staining. Representative images of hematoxylin and eosin, periodic acid–Schiff (PAS), and Masson trichrome staining of the renal cortex.

**Figure 6 ijms-23-08915-f006:**
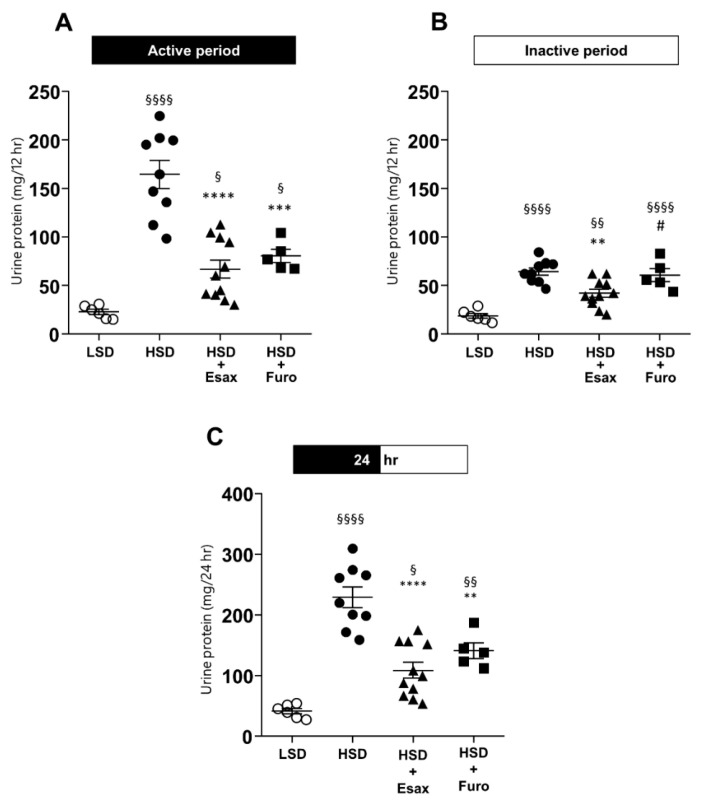
Urinary protein excretion. Mean 12 h urinary protein excretion in the (**A**) active and (**B**) inactive periods, and (**C**) at 24 h. ^§^
*p* < 0.05, ^§§^
*p* < 0.01, ^§§§§^
*p* < 0.0001 vs. LSD; ** *p* < 0.01, *** *p* < 0.001, **** *p* < 0.000 1 vs. HSD; ^#^
*p* < 0.05 vs. HSD plus esaxerenone.

## Data Availability

The data presented in this study are available in this article or Appendix A.

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
