# Peer review of "Association of Antihypertensive Effects of Esaxerenone with the Internal Sodium Balance in Dahl Salt-Sensitive Hypertensive Rats"

_ijms, 2022, doi:10.3390/ijms23168915_

Round 1
Reviewer 1 Report
In this manuscript, Hattori et al. studied the antihypertensive effects of esaxerenone in Dahl salt-sensitive hypertensive rats. They found that the treatment with esaxeronone, a nonsteroidal mineralocorticoid receptor blocker reduced the blood pressure in both a non-dipper and dipper pattern. On the contrary, furosemide was able to significantly reduce Blood pressure only in a dipper pattern. Moreover, esaxerenone treatment attenuated glomerulosclerosis, tubulointerstitial fibrosis, and urinary protein excretion induced by high salt loading. Based on these observations, the authors concluded that esaxerenone treatment-induced reduction of blood pressure and renoprotection are associated with body sodium homeostasis.
Major comments
1. the manuscript is difficult to understand since in figures 1A and 1B, and 4, there is no statistically significant difference between LSD, HSD and HSD+treatments.
2. Since there is no significant difference, it is difficult to understand that esaxerenone treatment decreases the sodium balance in the active period.
3. The authors must be measured the level of renin and angiotensin II as well as the expression of ACE, ACE2, AT-1, and AT-2 receptors.
4. the authors have any data about the comparison between esaxerenone treatment and valsartan treatment in HSD?
5. the authors should compare the effect of esaxerenone with another nonsteroidal mineralocorticoid antagonist and not with the furosemide. The authors should be clarified this choice.
Minor comments
In Drugs and modified chow, the authors should be written the exact dose of esaxerenone and furosemide.
Author Response
Responses to the comments raised by reviewer #1
General comment
In this manuscript, Hattori et al. studied the antihypertensive effects of esaxerenone in Dahl salt-sensitive hypertensive rats. They found that the treatment with esaxeronone, a nonsteroidal mineralocorticoid receptor blocker reduced the blood pressure in both a non-dipper and dipper pattern. On the contrary, furosemide was able to significantly reduce Blood pressure only in a dipper pattern. Moreover, esaxerenone treatment attenuated glomerulosclerosis, tubulointerstitial fibrosis, and urinary protein excretion induced by high salt loading. Based on these observations, the authors concluded that esaxerenone treatment-induced reduction of blood pressure and renoprotection are associated with body sodium homeostasis.
Response
We thank the reviewer for the useful comments, which have helped us to improve our manuscript. Because we were only allowed 10 days to revise our manuscript, we were not able to conduct substantial additional animal experiments. However, we have made extensive changes in the manuscript to address all issues raised by the reviewer, as detailed in our point-by-point responses below.
Major Comments
Comment-1
The manuscript is difficult to understand since in figures 1A and 1B, and 4, there is no statistically significant difference between LSD, HSD and HSD+treatments.
Response
We thank the reviewer for this thoughtful comment. We sincerely apologize for the complicated figures. We have attempted to make the figures as easy to understand as possible. However, because we needed to include a large amount of data from many animal experiments, these figures are a little difficult to understand.
Because we focused on averaged 12 h data in the active and inactive periods, we did not show statistical analysis for every hour of data in the original manuscript. However, in response to the reviewer’s suggestion, an additional analysis has been performed to analyze every hour of data in Figure 1A and 1B in the revised manuscript. Our data showed that, at most of the time points, there was a significant difference in systolic and mean arterial pressure between low- and high-salt-loaded rats. Furthermore, esaxerenone-treated HSD-fed DSS rats showed a significantly lower blood pressure than HSD-fed rats at certain time points during the active and inactive periods. As shown in Figure 4, although we were not able to observe a significant difference, the sodium balance appeared to decrease in the active period. Importantly, a significant difference in the sodium balance at 24 h was observed in esaxerenone-treated HSD-fed DSS rats compared with HSD-fed rats at week 4. To avoid any confusion, we have revised the descriptions in the revised manuscript as follows.
Results (page 3, lines 102-110)
“HSD-fed DSS rats had significantly higher hourly measurements (most time points) and averaged 24-h mean systolic arterial pressure ([SAP] 215±4 mmHg), mean diastolic arterial pressure ([DAP] 150±2 mmHg), and mean arterial pressure ([MAP] 181±5 mmHg) than LSD-fed DSS rats with feeding for 4 weeks (Figure 1A, B and Figure S1A). However, concomitant treatment with esaxerenone significantly suppressed this elevation in hourly measurements at certain time points and averaged 24-h SAP (17±7 mmHg), DAP (122±5 mmHg), and MAP (149±6 mmHg). The supplementation of an HSD with furosemide tended to reduce SAP (204±8 mmHg), DAP (140±7 mmHg), and MAP (170±7 mmHg), but these changes were not significant.”
Results (page 7, lines 182-188)
“However, in the subsequent weeks, the sodium balance was similar in HSD-fed DSS rats and HSD-fed plus furosemide-treated rats, but the sodium balance appeared to show a downward trend in HSD-fed plus esaxerenone-treated rats in the active period (Figure 4A). Because food intake was low during the inactive period, the sodium balance was not obviously different among the groups (Figure 4B). Importantly, the 24-h sodium balance was significantly lower in HSD-fed plus esaxerenone-treated rats at week 4 (Figure 4C).”
Comment-2
Since there is no significant difference, it is difficult to understand that esaxerenone treatment decreases the sodium balance in the active period.
Response
We agree with the reviewer’s comment that a significant difference in the sodium balance was not observed during the active period. However, there appeared to be a clear downward trend with esaxerenone. We believe that the reason for the lack of a significant difference may be due to the small number of rats used. Because of the large number of animal protocols conducted in this study, the Kagawa University Animal Committee did not approve the use of more animals, for animal welfare reasons. We have added the issue of the small number of rats as a limitation to this study.
Discussion (page 9, lines 278-281)
“The sodium balance tended to be reduced, but was not significant, in the active period. However the 24-h sodium balance was significantly lower in esaxerenone-treated rats. A limitation of this study is that a relatively small number of rats was used for measuring the sodium balance.”
Comment-3
The authors must be measured the level of renin and angiotensin II as well as the expression of ACE, ACE2, AT-1, and AT-2 receptors.
Response
We thank the reviewer for this important suggestion. Because our research team continues to lead the world in the study of the renin–angiotensin system (e.g., Pharmacol Rev. 2007; 59(3): 251-87 and Pharmacol Rev. 2022; 74(3): 462-505), we agree with the reviewer’s comment that the renin–angiotensin system should be investigated. Although this study did not aim to investigate the renin–angiotensin system, our previous studies already measured plasma renin activity and the intrarenal renin–angiotensin system in detail (Li et al. Hypertens Res. 2019 Jun;42(6):769-778).
Comment-4
The authors have any data about the comparison between esaxerenone treatment and valsartan treatment in HSD?
Response
In our previous study (Li et al. Hypertens Res. 2019 Jun;42(6):769-778), we already compared esaxerenone with losartan in high salt-loaded DSS rats.
Comment-5
The authors should compare the effect of esaxerenone with another nonsteroidal mineralocorticoid antagonist and not with the furosemide. The authors should be clarified this choice.
Response
As the reviewer may be aware, another nonsteroidal mineralocorticoid antagonist, finerenone, has recently become available for clinical use (N Engl J Med. 2020; 383(23): 2219-2229 and N Engl J Med. 2021; 385(24): 2252-2263). However, unfortunately, finerenone is not available yet for use in basic research without complicated negotiations with the company (Bayer Co., Germany). Nevertheless, comparing the effects of two nonsteroidal mineralocorticoid antagonists is important, and we would like to compare them in future experiments. Because we believe that this is an important issue, related discussion has been added to the revised manuscript as follows.
Discussion (page 9, lines 261-263)
“Moreover, finerenone, which is another non-steroidal MRB, dose dependently showed natriuretic potency in healthy volunteers with a fludrocortisone challenge [29].”
Minor comments
In Drugs and modified chow, the authors should be written the exact dose of esaxerenone and furosemide.
Response
We provided the drugs with the chow, and the intake of chow varied among the rats day by day. Therefore, calculating the exact dose of drugs in this experimental setting is difficult. Except when using metabolic cages, several rats were often kept in the same cage in accordance with the guidance of the animal committee. Therefore, individual food intake could only be roughly measured. We have reviewed previous papers on similar basic experiments in rats and mice in which MRBs were mixed with food and administered, and the drug doses were described in the same way as we explained in the Method section. Furthermore, we consulted with the animal committee of Kagawa University to be sure of this procedure, and they confirmed that there is no problem with this method of administration from an animal ethics standpoint. Notably, doses of esaxerenone and furosemide have been determined on the basis of previous studies in rats (Li et al. Hypertens Res. 2019 Jun;42(6):769-778 and Yoshida et al. Cardiovasc. Res. 2005, 68, 118-127).
We would appreciate your approval of the method of administering the drug to be mixed with the food in this study, which we performed on the basis of previous literature and advice from the animal committee of Kagawa University.

Reviewer 2 Report
Hattori et al. showed that treatment with esaxerenone but not furosemide suppressed the HSD-induced increase in BP and changed the diurnal variation of BP from the non-dipper to the dipper pattern in Dahl salt-sensitive (DSS) rat. However, some of the results are not fully interpretable and would need to be examined.
1) In this paper, esaxerenone significantly suppressed HSD-induced hypertension in the active periods and more strongly in the inactive periods, while furosemide did not suppress it in either periods. Therefore, the authors said that Esaxerenone improves renal function by changing the non-dipper to dipper pattern in BP. However, Furosemide significantly improves proteinuria during active periods as Esaxerenone. Is there a difference in total amounts of daily proteinuria between Esaxerenone and Furosemide? Sufficient data are needed to determine whether the dipper to non-dipper type shift in blood pressure with esaxerenone treatment contributes to improved renal function. 
2) In this article, the authors said that the esaxerenone-induced increase in the urinary sodium to potassium ratio contributes to the improvement in diurnal variation of BP from the non-dipper to dipper pattern in HS-loaded DSS rats.
Furosemide inhibits NaCl and K+ reabsorption by suppressing the Na+, K+, 2Cl- co-transport carrier (NKCC2) on the luminal side membrane of the thick ascending limb of the Henle's loop, but because of its short duration of action, the effect of furosemide is considered to be smaller during the inactive period when no food is ingested, resulting in a marked decrease in urinary K excretion.
Therefore, the Na/K ratio would be higher in the HSD+Esaxerenone group than in the HSD+Furosemide group during active periods, but the difference between the two groups would disappear during inactive periods. However, Furosemide does not suppress the HSD-induced increase in blood pressure during the inactive periods. Therefore, Na/K may be unrelated to the mechanism of action of esaxerenone, which reduces blood pressure more strongly during inactive as well as active periods.
3)In supplement data Figure S2 F, there is an extra asterisk near the black dot on D5.
Author Response
Responses to the comments raised by reviewer #2
General comment
Hattori et al. showed that treatment with esaxerenone but not furosemide suppressed the HSD-induced increase in BP and changed the diurnal variation of BP from the non-dipper to the dipper pattern in Dahl salt-sensitive (DSS) rat. However, some of the results are not fully interpretable and would need to be examined.
Response
We thank the reviewer for the useful comments, which have helped us to improve our manuscript. Because we were only allowed 10 days to revise our manuscript, we were not able to conduct substantial additional animal experiments. However, we have made extensive changes in the manuscript to address all issues raised by the reviewer, as detailed in our point-by-point responses below.
Comment-1
In this paper, esaxerenone significantly suppressed HSD-induced hypertension in the active periods and more strongly in the inactive periods, while furosemide did not suppress it in either periods. Therefore, the authors said that Esaxerenone improves renal function by changing the non-dipper to dipper pattern in BP. However, Furosemide significantly improves proteinuria during active periods as Esaxerenone. Is there a difference in total amounts of daily proteinuria between Esaxerenone and Furosemide? Sufficient data are needed to determine whether the dipper to non-dipper type shift in blood pressure with esaxerenone treatment contributes to improved renal function. 
Response
We thank the reviewer for the excellent suggestions. We agree with the reviewer’s comment that our data are not sufficient to conclude that the shift in the dipping pattern of blood pressure was associated with an improvement in renal function. Even though proteinuria was significantly reduced in the esaxerenone and furosemide treatment groups compared with the vehicle-treated group, there was no significant difference in proteinuria between these two treatment groups. In response to the reviewer’s suggestions, we have revised the text to avoid any descriptions regarding any relationship of a diurnal variation in blood pressure with renal function.
Discussion (page 10, lines 301-302)
“These data suggest that esaxerenone-induced improvement of renal histological changes and reduced proteinuria are associated with a reduction in BP.”
Comment-2
In this article, the authors said that the esaxerenone-induced increase in the urinary sodium to potassium ratio contributes to the improvement in diurnal variation of BP from the non-dipper to dipper pattern in HS-loaded DSS rats. Furosemide inhibits NaCl and K+ reabsorption by suppressing the Na+, K+, 2Cl- co-transport carrier (NKCC2) on the luminal side membrane of the thick ascending limb of the Henle's loop, but because of its short duration of action, the effect of furosemide is considered to be smaller during the inactive period when no food is ingested, resulting in a marked decrease in urinary K excretion. Therefore, the Na/K ratio would be higher in the HSD+Esaxerenone group than in the HSD+Furosemide group during active periods, but the difference between the two groups would disappear during inactive periods. However, Furosemide does not suppress the HSD-induced increase in blood pressure during the inactive periods. Therefore, Na/K may be unrelated to the mechanism of action of esaxerenone, which reduces blood pressure more strongly during inactive as well as active periods.
Response
We understand and agree with the reviewer’s important comments. Therefore, we have revised descriptions by removing any speculation suggesting an association of a shift in the dipping pattern of blood pressure with the Na/K ratio in the Discussion section.
Comment-3
In supplement data Figure S2F, there is an extra asterisk near the black dot on D5.
Response
We have deleted the extra asterisk.

Reviewer 3 Report
Authors presented results of interesting study about association of antihypertensive effects of esaxerenone with the internal sodium balance in Dahl salt-sensitive hypertensive rats. I have some comments:
1) Abstract should be structured: Background, Objective, Methods, Results, Conclusions.
2) Supplement the descriptive statistics with the amount of variation in the parameters. Given the very small number of experimental objects, it is reasonable to show min-max.
Author Response
Responses to the comments raised by reviewer #3
General comment
Authors presented results of interesting study about association of antihypertensive effects of esaxerenone with the internal sodium balance in Dahl salt-sensitive hypertensive rats. I have some comments:
Response
We thank the reviewer for the useful comments, which have helped us to improve our manuscript. In response to the concerns and issues raised by the reviewer, we have made extensive changes in the manuscript, as detailed by our point-by-point responses below.
Comment-1
Abstract should be structured: Background, Objective, Methods, Results, Conclusions.
Response
In accordance with the reviewer’s comment, we have structured the abstract as suggested.
Comment-2
Supplement the descriptive statistics with the amount of variation in the parameters. Given the very small number of experimental objects, it is reasonable to show min-max.
Response
We thank the reviewer for this suggestion. We have mentioned the raw data with standard error values for the most important parameters in the text. With regard to the cross-sectional, one-factor data that we have already shown, the individual data for each experimental group have been provided to understand the range of the data.

Round 2
Reviewer 1 Report
The authors have explained all my doubts
Author Response
Responses to the comments raised by reviewer #1
The authors have explained all my doubts.
Response
We thank the reviewer for the useful comments, which have helped us to improve our manuscript.

Reviewer 2 Report
The authors have made improvements to the article, but still have the following problems.
(1) The authors noted below, but the higher urinary potassium excretion in the HSD plus furosemide group than in the HSD plus esaxerenone group seems to affect the higher Na/K ratio only at active period, and there is no difference in Na/K ratio at inactive period between the two groups.
P4 L136
Notably, urinary potassium excretion was relatively higher in HSD-fed plus furosemide-treated rats than in HSD-fed plus esaxerenone-treated rats (Figure S2C.D).
Therefore, the urinary sodium to potassium ratio was greater in HSD-fed plus esaxerenone-treated rats than in HSD-fed rats or HSD-fed plus furosemide-treated rats in the active and inactive periods (Figure S2E.F)
(2) Regarding the following statement about Figure3, was there a significant difference between HSD-fed DSS and LSD-fed DSS in sodium concentration of skin or skinned carcass and whole carcass? And was there a significant difference between HSD plus furosemide and HSD plus esaxerenone in sodium concentration of skin or skinned carcass and whole carcass?
If there are significant differences in these, please enter a symbol on the graph indicating a significant difference and explain it in the legend. If there is no significant difference, this should be clearly stated in the text.
P6 L162
As expected, HSD-fed DSS rats showed higher sodium concentrations in the skin, skinned carcass, and whole carcass than those in LSD-fed rats (Figure 3A, D, G).
HSD-fed plus furosemide-treated rats showed higher sodium concentrations, while HSD-fed plus esaxerenone-treated rats showed lower sodium concentrations in the skin, skinned carcass, and whole carcass.
(3) As I pointed out in my previous comment, the explanation in Figure 6 gives a misleading impression that esaxerenone has a more pronounced inhibitory effect on proteinuria than furosemide. If there is no difference in total daily proteinuria between HSD plus esaxerenone and HSD plus furosemide, this should be stated in the text or shown in the graph. Figure 6 shows that furosemide as well as esaxerenone significantly suppressed the increase in proteinuria induced by HSD at active period, with no difference between the two groups.
The loss of the proteinuria suppression effect of furosemide at the inactive period may be due to a marked decrease in the intake of furosemide and esaxerenone in the diet, resulting in the loss of only the effect of furosemide, which has a short duration of action, and this should be discussed as the limitation.
(4) The following description in abstract appears inappropriate.
P1 L34
However, a significant increase in the sodium/potassium ratio was only observed in esaxerenone-treated rats.
Author Response
Responses to the comments raised by reviewer #2
General comment
The authors have made improvements to the article, but still have the following problems.
Response
We thank the reviewer for the useful comment, which have helped us to improve our manuscript and strengthened the implications of our results.
Comment-1
The authors noted below, but the higher urinary potassium excretion in the HSD plus furosemide group than in the HSD plus esaxerenone group seems to affect the higher Na/K ratio only at active period, and there is no difference in Na/K ratio at inactive period between the two groups.
P4 L136
Notably, urinary potassium excretion was relatively higher in HSD-fed plus furosemide-treated rats than in HSD-fed plus esaxerenone-treated rats (Figure S2C.D).
Therefore, the urinary sodium to potassium ratio was greater in HSD-fed plus esaxerenone-treated rats than in HSD-fed rats or HSD-fed plus furosemide-treated rats in the active and inactive periods (Figure S2E.F) 
Response
We thank the reviewer for thoughtful comments. We totally agree with the reviewer’ comments that the bioavailability of furosemide and esaxerenone is different, and it is possible that the duration of action of furosemide affect sodium/potassium ratio. To avoid any confusion, we have removed all the data and discussion related to the urinary sodium/potassium ratio in the revised manuscript.
Comment-2
Regarding the following statement about Figure3, was there a significant difference between HSD-fed DSS and LSD-fed DSS in sodium concentration of skin or skinned carcass and whole carcass? And was there a significant difference between HSD plus furosemide and HSD plus esaxerenone in sodium concentration of skin or skinned carcass and whole carcass?
If there are significant differences in these, please enter a symbol on the graph indicating a significant difference and explain it in the legend. If there is no significant difference, this should be clearly stated in the text.
P6 L162
As expected, HSD-fed DSS rats showed higher sodium concentrations in the skin, skinned carcass, and whole carcass than those in LSD-fed rats (Figure 3A, D, G).
HSD-fed plus furosemide-treated rats showed higher sodium concentrations, while HSD-fed plus esaxerenone-treated rats showed lower sodium concentrations in the skin, skinned carcass, and whole carcass.
Response
We appreciate the reviewer for indicating these issues. We have revised manuscript according to your instructions. In the revised manuscript, we have re-analyzed to compare the mean of each group with the mean of every other groups. Based on this multiple comparison, we have added the symbol in the graph and revised the text, as follows.
Results (page 6, lines 164-172)
As expected, HSD-fed DSS rats showed significantly higher sodium concentrations in the skin, skinned carcass, and whole carcass than those in LSD-fed rats (Figure 3A, D, G). HSD-fed plus furosemide-treated rats showed significantly higher sodium concentrations in skin compared to the HSD-fed rats, while HSD-fed plus esaxerenone-treated rats showed a reduced trend of sodium concentrations (significantly higher than LSD but not with HSD) in the skin, skinned carcass, and whole carcass. Importantly, level of sodium in the skinned and whole carcass was significantly higher in furosemide-treated rats compared with the esaxerenone-treated rats.
Comment-3
As I pointed out in my previous comment, the explanation in Figure 6 gives a misleading impression that esaxerenone has a more pronounced inhibitory effect on proteinuria than furosemide. If there is no difference in total daily proteinuria between HSD plus esaxerenone and HSD plus furosemide, this should be stated in the text or shown in the graph. Figure 6 shows that furosemide as well as esaxerenone significantly suppressed the increase in proteinuria induced by HSD at active period, with no difference between the two groups.
The loss of the proteinuria suppression effect of furosemide at the inactive period may be due to a marked decrease in the intake of furosemide and esaxerenone in the diet, resulting in the loss of only the effect of furosemide, which has a short duration of action, and this should be discussed as the limitation.
Response
We apologize for the incomplete response in the first revision. We do agree with your comment that bioavailability is different among the furosemide and esaxerenone, and we also think it is a limitation for this study. In accordance with your suggestions, we have added the following discussion as a weak point of this study.
Discussion (page 10, lines 310-313)
“In contrast, furosemide treatment reduced urinary protein excretion in the active but not inactive period compared to the HSD-fed rats. This might be due to the short duration of action that seems a limitation for this study.”
Comment-4
The following description in abstract appears inappropriate.
P1 L34
However, a significant increase in the sodium/potassium ratio was only observed in esaxerenone-treated rats.
Response
Thank you very much for your kind suggestion. We have deleted the line from the abstract to avoid any misunderstanding.
